## TOPICAL REVIEW

# Pacemaking in the heart: a redundant and robust system of mutually entrained oscillators driving cardiac automaticity

Eilidh A. MacDonald[1] 🆔 and T. Alexander Quinn[2,3] 🆔

[1] *School of Cardiovascular and Metabolic Health, University of Glasgow, Glasgow, UK*
[2] *Department of Physiology and Biophysics, Dalhousie University, Halifax, Canada*
[3] *School of Biomedical Engineering, Dalhousie University, Halifax, Canada*

Handling Editors: Laura Bennet & Bernard Drumm

The peer review history is available in the Supporting information section of this article (https://doi.org/10.1113/JP284757#support-information-section).

**Eilidh A. MacDonald** is a Postdoctoral Research Fellow in the School of Cardiovascular & Metabolic Health at the University of Glasgow. Her research is focused on mechanical determinants of sinoatrial node function and drivers of adverse cardiac remodelling that lead to heart failure and arrhythmias. **T. Alexander Quinn** is the Director of the Cardiac Autoregulation & Arrhythmias Laboratory and a Professor of Physiology & Biophysics and Biomedical Engineering at Dalhousie University. His research is focused on understanding the intrinsic regulation of cardiac function and the contribution of auto-regulatory mechanisms to deadly arrhythmias in cardiovascular disease and ageing.

**Abstract figure legend** Cardiac pacemaking: the redundant and robust organ-to-subcellular system driving the spontaneous electrical activity responsible for regular, rhythmic excitation of the heart.

**Abstract** The heart is an electrically controlled, mechanical pump that provides a constant supply of blood to the body. It is not surprising, then, that the heartbeat, which occurs on average one or twice every second and over 2.5 billion times without fail in most lifetimes, is maintained by a redundant and robust pacemaking system that ensures regular, rhythmic cardiac excitation. This excitation is initiated in the sinoatrial node (the heart's natural pacemaker), which displays intrinsic spontaneous electrical activity, responsible for the heart's automaticity. As part of the special issue on Pacemaking in Multicellular Organ Systems, in this review we provide a basic overview of the mechanisms responsible for cardiac automaticity intended for a general audience – to allow for comparison with other organs in which pacemaking activity is present – and discuss often overlooked factors critical for integrated cardiac pacemaker function. Ultimately, we hope that a better understanding of pacemaking in the heart, and how it relates to that seen in other organs, will improve our overall understanding of physiological cardiac function, automaticity observed in experimental model systems, and aberrant excitation responsible for deadly cardiac arrhythmias.

(Received 1 December 2025; accepted after revision 13 April 2026; first published online 28 April 2026)

**Corresponding author** T. Alexander Quinn, Department of Physiology and Biophysics, Dalhousie University, Halifax, NS B3H 4R2, Canada. Email: alex.quinn@dal.ca

## Extraordinarily spontaneous: cardiac automaticity and pacemaking

When the heart is removed from the body, it continues to beat. Similarly, some freshly isolated cardiac tissues or single cardiomyocytes will spontaneously contract, as often do cultures of neonatal or induced pluripotent stem cell-derived cardiomyocytes (iPSC-CM). For those of us working with hearts, cardiac tissues, or cell preparations, this is often unquestioned and potentially underappreciated. However, when someone from outside of the cardiovascular field observes a spontaneously beating isolated heart, or a dish of spontaneously contracting tissue or cells, they are often astonished. And rightly so – the spontaneous nature of cardiac activity, often referred to as automaticity, is fascinating – and essential for life.

In the healthy heart, the electrical excitation that initiates each heartbeat, which is commonly referred to as 'pacemaking', begins within the organ itself, in a region of tissue known as the sinoatrial node (SAN). In the 140 years since the intrinsic nature of cardiac excitation was identified (Gaskell, 1882), an entire field of research investigating cardiac automaticity has emerged. From this, we have gained incredible insight into the physiology of pacemaker function, including basic ionic mechanisms involved, sources of its regulation, and causes of its dysfunction. Furthermore, it has been recognised that automaticity occurs not only in the SAN, but also in other cardiac tissues such as the atrioventricular node and Purkinje fibres, resulting in a well-accepted 'textbook' understanding of cardiac pacemaking (Bartos et al., 2015; Boyett et al., 2000; Irisawa et al., 1993; Lakatta & DiFrancesco, 2009; Lakatta et al., 2010; MacDonald et al., 2020; Mangoni & Nargeot, 2008; Opthof, 1988). As the above list of papers from leaders in the field attest, our prevailing understanding of pacemaking in the heart has been extensively reviewed for the cardiac expert. Here, our goal is to provide a general outline of the mechanisms driving cardiac automaticity for the uninitiated, to allow non-cardiac researchers to contrast cardiac pacemaking activity with that found in other organs, as reviewed in the other papers in this special issue focused on Pacemaking in Multicellular Organ Systems. In the process, we also extend the discussion of cardiac automaticity to important but less considered factors (such as the importance of tissue-level electrophysiological and mechanical heterogeneity and pacemaker entrainment by mechano-electric feedback), to create space for emerging evidence and novel ideas regarding pacemaking in the heart.

## Keeping the rhythm: the ionic basis for cardiac automaticity

A primary question has dominated the field of cardiac pacemaker research: what drives the automaticity of the SAN and other pacemaker tissue? The specific ionic mechanisms that result in spontaneous SAN excitation have been intensely studied, yet highly contested (Difrancesco & Noble, 2012; Lakatta & DiFrancesco,

2009; Maltsev & Lakatta, 2012). Ultimately, two sides have emerged: one asserting that SAN automaticity is driven primarily by transmembrane currents, which comprise a system known as the 'membrane-clock', with the other alleging it is driven primarily by a system of intracellular calcium ($Ca^{2+}$) cycling, known as the '$Ca^{2+}$-clock' (Lakatta & DiFrancesco, 2009).

To whichever side one prescribes (or otherwise), it is clear that spontaneous SAN excitation is the result of the inward flux of positive ions across the cell membrane during the period of cardiac rest known as 'diastole'. In working cardiomyocytes (found in the myocardium of the atria and ventricles), membrane potential is held at a stable negative value in diastole, until depolarised by a neighbouring cell or external electrical stimulus. In pacemaker cells, however, there is a gradual depolarisation of membrane potential during this period, often referred to as spontaneous diastolic depolarisation

(SDD), commencing at the most negative membrane potential of the cell (maximum diastolic potential, MDP) and continuing until it reaches the threshold for action potential generation (Bartos et al., 2015). Critically, the rate of SDD, along with the MDP and threshold potential are key determinants of the frequency of SAN excitation, and thus heart rate (Mangoni & Nargeot, 2008).

In the SAN, the inward flux of positive charge leading to SDD and cardiac excitation comes from multiple sources (Fig. 1A). The primary ionic current included in the membrane-clock is the 'funny' current ($I_f$), an inward flow of cations that passes through hyperpolarisation-activated, cyclic nucleotide-gated (HCN) channels (Difrancesco, 2010). Unlike most voltage-gated channels, which open when membrane potential becomes more positive, HCN channels are activated by increasingly negative membrane potential, such that as the cell repolarises more HCN channels

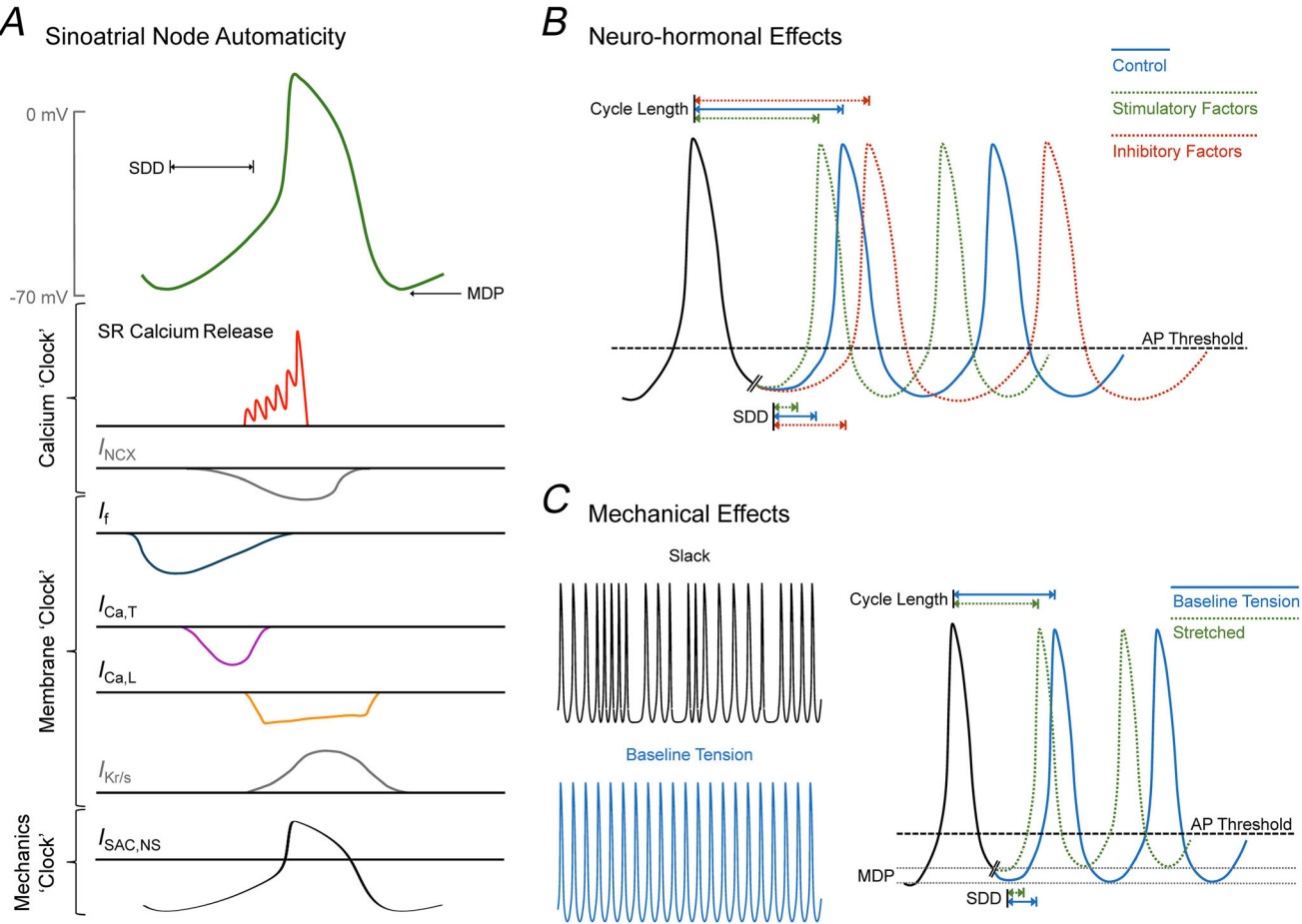

**Figure 1. Cellular mechanisms of spontaneous sinoatrial node excitation**
*A*, sinoatrial node action potential and ionic fluxes responsible for spontaneous diastolic depolarisation (SDD) and cardiac automaticity. Neurohormonal (*B*) and mechanical (*C*) effects are shown on the sinoatrial node action potential, SDD and cycle length, resulting in a change in heart rate and the regularity of sinoatrial node excitation. $I_{Ca,L}$, L-type calcium current; $I_{Ca,T}$, T-type calcium current; $I_f$, 'funny' current; $I_{Kr/s}$, rapid/slow delayed rectifier potassium current; $I_{NCX}$, sodium-calcium exchanger current; $I_{SAC,NS}$, cation non-selective stretch-activated channel current; MDP, maximum diastolic potential; SR, sarcoplasmic reticulum.

open, causing depolarisation – an unusual behaviour that gives the 'funny' current its name. Thus, $I_f$ contributes to the early period of SDD by opposing membrane repolarisation (Difrancesco, 2010). As membrane potential increases during SDD, there is a decrease in $I_f$, but, at the same time, inward $Ca^{2+}$ currents become activated; first the transient T-type $Ca^{2+}$ current ($I_{Ca,T}$; passing through Cav3.1 channels), which activates and inactivates relatively quickly, and then, at a slightly higher membrane potential, the long-lasting L-type $Ca^{2+}$ current ($I_{Ca,L}$; initially passing through Cav1.3 channels and then Cav1.2 channels, which are ultimately responsible for the AP upstroke) (Mesirca et al., 2015). Additionally, at least in some species, there may be a contribution of other ionic currents to SDD, such as those passing through fast sodium ($Na^+$; Nav1.5) channels (Lei et al., 2005) or transient receptor potential-canonical (TRPC) channels (Ju et al., 2007), which have been shown to affect excitation rate and be heterogeneously (albeit in most cases minimally) expressed in the SAN.

The contribution of the $Ca^{2+}$-clock to SDD and spontaneous excitation, on the other hand, begins with spontaneous or triggered localised release of $Ca^{2+}$ from the sarcoplasmic reticulum (SR) via ryanodine receptors (RyR) (Lakatta et al., 2008). Although much of this $Ca^{2+}$ is put back into the SR by the sarco/endoplasmic reticulum $Ca^{2+}$-ATPase (SERCA), a portion is extruded from the cell by the $Na^+$-$Ca^{2+}$ exchanger, through which one $Ca^{2+}$ ion leaves the cell as three $Na^+$ ions enter the cell, resulting in a net positive influx of charge that causes membrane depolarisation (Lakatta et al., 2008).

Of course, repolarising ionic currents are also fundamental to cardiac pacemaker activity because pacemaker cells must 'reset' with each heartbeat. Indeed, prior to the identification of the membrane- and $Ca^{2+}$-clock ionic fluxes responsible for SDD, decay of repolarising potassium ($K^+$) currents at the end of the SAN action potential was considered by many to be the main driver of SAN automaticity. As in working cardiomyocytes, L-type $Ca^{2+}$ current inactivates near the end of the action potential, at which point the current passed by rapid ($I_{Kr}$) and slow ($I_{Ks}$) delayed rectifier $K^+$ channels increases, repolarising the SAN to its MDP (Aziz et al., 2018). As membrane potential becomes more negative and these repolarising currents decay, the simultaneous increase in $I_f$ begins to drive SDD. Importantly, unlike in working cardiomyocytes, in which current passed through inwardly rectifying $K^+$ channels ($I_{K1}$) maintains the stable negative membrane potential during diastole, the current is largely absent in SAN myocytes, so does not prevent SDD (Bartos et al., 2015). Additionally, other $K^+$ currents may contribute to SAN repolarisation in some species, such as that passing through ultra-rapid inwardly rectifying, ACh-activated, ATP-sensitive (Aziz et al., 2018), $Ca^{2+}$-activated (Weisbrod et al., 2016), or two-pore-domain (TREK-1) (Unudurthi et al., 2016) $K^+$ channels, as well as the outward flux of positive charge generated through the sodium-potassium pump ($Na^+$-$K^+$ ATPase) as it extrudes three $Na^+$ ions in exchange for two $K^+$ ions to help maintain electrochemical homeostasis (Sirenko et al., 2016). Importantly, the rate of repolarisation, along with SDD slope, MDP and threshold potential, also contributes to the overall frequency of SAN excitation and the resulting heart rate.

Importantly, the above mechanisms involved in SAN automaticity are highly influenced by neurohormonal factors, allowing for tight central and local control of pacemaker activity and the adaptation of heart rate to changing physiological demands (Fig. 1B). Neurohormonally altered transmembrane ionic flux generally results from activation of intracellular signalling pathways by ligand stimulation of receptors on SAN cells, *via* peptides released by nerves of the autonomic nervous system and by circulating or locally released peptides and hormones (MacDonald et al., 2020). For example, catecholamines released by nerves of the sympathetic branch of the autonomic nervous system bind to adrenergic receptors on SAN cells, activating downstream pathways that increase the inward flux of positive charge, increasing MDP and the rate of SDD, and thus SAN excitation rate. Conversely, ACh released by parasympathetic autonomic nerves binds to cholinergic receptors, driving intracellular signalling that reduces cation influx and activating ACh-activated $K^+$ channels, which increases K+ efflux, thus reducing MDP, SDD slope and repolarisation rate, and ultimately the frequency of SAN excitation (Gordan et al., 2015; Zheng et al., 2026). Similarly, various endocrine, paracrine, and autocrine hormones and peptides affect ionic flux in SAN cells, altering MDP and the rate of SDD and repolarisation, and ultimately heart rate (Beaulieu & Lambert, 1998).

The above is only a general overview of the mechanisms and regulation of cardiac pacemaker activity. Greater detail is available in the many comprehensive reviews on the subject (Bartos et al., 2015; Boyett et al., 2000; Irisawa et al., 1993; Lakatta & DiFrancesco, 2009; Lakatta et al., 2010; MacDonald et al., 2020; Mangoni & Nargeot, 2008; Opthof, 1988).

## Mechanics matters: an additional 'clock' joins the mix

Although the two 'clocks' assumed to be responsible for SDD, combined with the currents critical for repolarisation, can largely account for cardiac automaticity and pacemaker activity in the heart, it has been clear for over 100 years that there is another important contributor to the maintenance of cardiac rhythm. During the diastolic period of each heartbeat, the heart fills with blood, followed by its emptying

during contraction, causing an oscillating cycle of myocardial stretch and shortening, including in the SAN and other pacemaker tissues. In the SAN, the response to stretch is a magnitude-dependent increase in the rate of excitation, known as the 'Bainbridge response' (Quinn & Kohl, 2022). This mechanically-induced response is considered to occur through activation of cation non-selective stretch-activated channels ($SAC_{NS}$) and mechano-sensitivity of membrane- or $Ca^{2+}$-clock components (Quinn & Kohl, 2021), thus allowing for adaptation of heart rate to changes in mechanical load (Quinn & Kohl, 2012). SAN mechano-sensitivity may also be a critical contributor to normal pacemaker function because, in unloaded isolated SAN preparations, the beating rate is often irregular, possibly as a result of the failure of other pacemaker mechanisms to sustain a consistent SDD (Kim et al., 2018). In this case, it has been shown that application of a baseline level of diastolic stretch results in the stabilisation of rhythm (Stoyek et al., 2022), as is often unknowingly applied when experimentalists using isolated SAN or atria pin out the tissue, applying 'just enough' stretch to establish regularly occurring excitation. Beyond baseline tension, in the whole heart peak levels of SAN stretch during atrial filling will coincide with the period of SDD, such that stretch-induced depolarising currents will contribute to SDD, mechanically 'priming' pacemaker cells for excitation (Cooper et al., 2000), and allow for the beat-by-beat adaptation of heart rate to changes in venous return that occur during normal activity such as exercise, altered posture, or respiration (Quinn & Kohl, 2012). Thus, the contribution of mechano-sensitivity to cardiac pacemaking may be considered another pacemaker system; the 'mechanics-clock' (MacDonald & Quinn, 2021) (Fig. 1C).

### Time's up: let's move away from clocks, coupled or otherwise

Although dividing the mechanisms driving cardiac pacemaker activity into individual 'clocks' (and other collections of currents) may be appealing, it is their combined actions that form the redundant and robust system critical for maintaining the heart's regular, rhythmic pacemaker activity, as the 'individual' clocks are tightly coupled and interdependent, with the action of one influencing the others (Rosen et al., 2012). For example, changes in transmembrane ion flux and membrane potential driven by the membrane- and mechanics-clock influence intracellular $Ca^{2+}$ balance and the local release of $Ca^{2+}$ from the SR, which are critical to the $Ca^{2+}$-clock, whereas local $Ca^{2+}$ release activates the $Na^+$-$Ca^{2+}$ exchanger, which is located in the cell membrane and whose activity alters membrane currents. Furthermore, it has been shown that the relative contribution of each clock for driving pacemaking varies between SAN cells (Monfredi et al., 2018) and the various clock components are affected by intracellular regulatory signalling that helps co-ordinate their activity (MacDonald et al., 2020). Importantly, this means that cardiac pacemakers – and the SAN in particular – continue to spontaneously excite even with the loss of individual molecular components, ensuring their operation under widely varying conditions, thus engendering the protective redundancy and robustness required for a system so critical to life (Irisawa et al., 1993).

Accordingly, despite contradictory perspectives regarding the critical importance of individual molecular components for pacemaker automaticity, it is the combined, out-of-phase activity of the various contributors that results in the rhythmic membrane potential oscillations required for spontaneous excitation (and none on their own can produce these oscillations). As such, the 'distinct' systems involved are better considered a redundant and robust system of mutually entrained oscillators (Noble et al., 2010) (Fig. 1A) and, although some hold strong to the 'clocks' concept (calling them 'coupled clocks') (Lakatta et al., 2010), considering them individually limits the potenital for an integrated understanding of cardiac pacemaker function.

### The cell is swell, but the tissue is the issue

It is important to recognise that much of our understanding of pacemaker automaticity has been developed through investigations in isolated cells. Yet, excitation of an individual pacemaker cell in well-coupled tissue will not result in excitation of the surrounding tissue, due to electrotonic coupling and source–sink imbalance (Boyett et al., 2000). A critical feature for successful SAN excitation and depolarisation of the surrounding atrial myocardium is its structural heterogeneity. Pacemaker cells in the SAN have diverse volumes, morphologies, and ionic current densities (Monfredi et al., 2018; Verheijck, Wessels et al., 1998), are interspersed with fibroblasts, and are embedded within a matrix of fibrous connective tissue (Bleeker et al., 1980; Boyett et al., 2000; Ho & Sánchez-Quintana, 2016; Monfredi et al., 2010; Opthof et al., 1987). In the central region of the SAN, this includes a dense arrangement of small, interwoven spindle-shaped pacemaker cells, embedded in a dense collagen network, whereas in the peripheral SAN, the cell arrangement is more ordered and fibrotically isolated from the adjacent atrial tissue, with only a few functional pathways for the 'exit' of SAN excitation (Bleeker et al., 1980; Fedorov et al., 2012). The resulting diminished intercellular connectivity within the SAN and at the periphery reduces the

electrotonic load between individual pacemaker cells and between SAN tissue and the surrounding atrial myocardium, resulting in a shift of the source–sink balance to promote successful internodal excitation and activation of the heart.

Although reduced intercellular connectivity will help overcome electrotonic load within the SAN, it will also reduce the spatial synchronicity of cellular activity, which may negatively impact SAN excitation. To promote the synchronous excitation of individual pacemaker cells across the SAN – similar to the molecular mechanisms driving cardiac automaticity – its cellular activity is mutually entrained (Jalife, 1984). Critical to this mutual entrainment is a phenomenon known as 'phase-resetting', by which a subthreshold (i.e. non-excitatory) depolarising current into spontaneously beating cells results in an increase or a decrease in their beating rate, depending on the timing of the stimulation within their electrical cycle (Anumonwo et al., 1991). Phase-resetting has been shown to occur in SAN tissue in response to an externally applied subthreshold electrical stimulus (Jalife & Antzelevitch, 1979; Sano et al., 1978) and to spatially entrain, and thus synchronise, its cellular activity (Verheijck, Wilders et al., 1998). *In vivo*, the subthreshold depolarisation needed for phase resetting to occur in the SAN may in part come from stretch-induced currents during diastolic atrial filling (MacDonald & Quinn, 2021). Although many SAN cells experience stretch-induced depolarisation during a similar period of SDD, cells that are not firing synchronously will experience it at a different point in their electrical

cycle. The stretch-induced depolarisation elicited by this 'out-of-phase' stretch may reset the electrical activity of the non-synchronous cells, aligning their excitation to the other cells in the SAN, which would reduce electrical heterogeneity and stabilise heart rhythm by preventing abnormally fast or arrhythmic groups of cells from overtaking SAN pacemaking (Abramovich-Sivan & Akselrod, 1999; Ushiyama & Brooks, 1977) (Fig. 2).

## What is the cardiac pacemaker, anyway? A tale of not-so-'specialised' cells

Although the SAN is considered the primary cardiac pacemaker, as already mentioned, other tissues in the heart also show automaticity (i.e. the atrioventricular node and Purkinje fibres), which are thought of as subsidiary (secondary and tertiary) pacemakers that can take over in the case of SAN failure, conferring further robustness to pacemaking in the heart (Bartos et al., 2015). Indeed, even within a given pacemaker region such as the SAN, where there is typically a leading site of excitation under normal conditions, physiological changes can cause this 'leading pacemaker site' to shift (Brennan et al., 2020). Along with altering beating rate, autonomic nervous system stimulation of the SAN results in a spatial shift of the initial point of excitation, with sympathetic stimulation causing a superior shift and parasympathetic an inferior shift, whose distance scales with the associated change in the rate of excitation (Boyett et al., 2000; Lang & Glukhov, 2021; Opthof, 1988). These shifts in the leading pacemaker

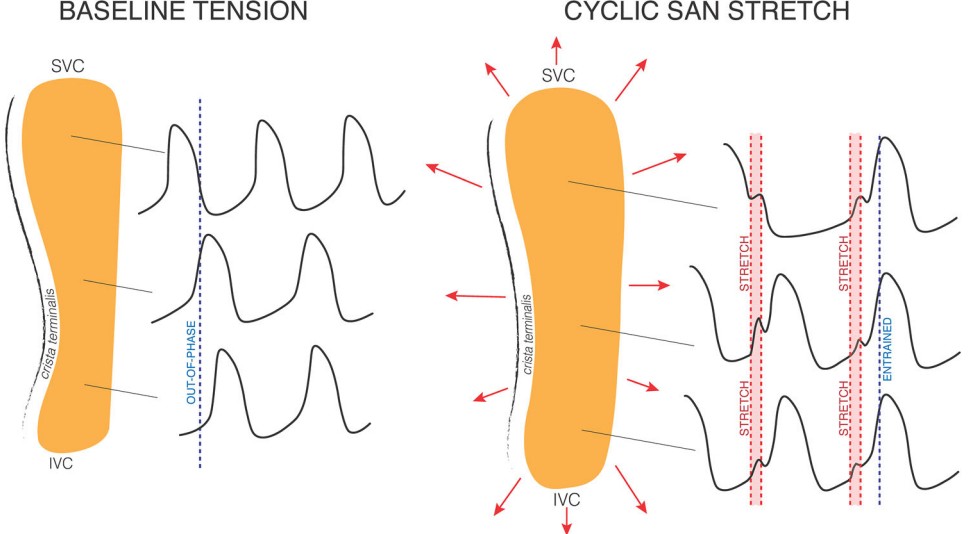

**Figure 2. Spatial mechanical entrainment of spontaneous sinoatrial node (SAN) excitation**
Under baseline tension, SAN excitation may be spatially asynchronous, as a result of structural and functional tissue heterogeneity and reduced intercellular connectivity needed to overcome electrotonic tissue effects. With cyclic stretch during diastolic atrial filling, mechanically-induced subthreshold depolarisations will lead to spatial entrainment of SAN excitation and reduced electrical heterogeneity via phase resetting. IVC, inferior vena cava; SVC, superior vena cava.

site have been attributed to regional heterogeneity in the balance of pacemaker currents in cells across the SAN and to the responsiveness of these cells to autonomic stimulation, either as a result of differences in local innervation or receptor density (Boyett et al., 2000; Lang & Glukhov, 2021; Opthof, 1988), and may provide even further safety to ensure consistent excitation of the heart under a variety of (patho-)physiological conditions.

The heterogeneity of pacemaker cells across the SAN (and other pacemaker tissue) reflects spatial differences in the degree of cellular differentiation. The cells in cardiac pacemaker tissues are often considered to be 'specialised' and – although they certainly are unique compared to cardiomyocytes throughout the rest of the heart – they more closely resemble the cells of the early embryonic heart, almost all of which show automaticity (Opthof, 2007). Accordingly, it may be that it is the cells of the working myocardium that have in fact differentiated and become 'specialised' (Gaskell himself considered the intrinsic rhythmicity of the heart to be arising from not fully differentiated cardiac tissue; Gaskell, 1882). This is clearly demonstrated (and often problematic) in experimental model systems such as cardiac cell culture and engineered heart tissue, where neonatal cells or iPSC-CM often show spontaneous activity because of a lack of adequate differentiation to working cardiomyocytes (Morad & Zhang, 2017). Similarly, subsidiary pacemakers in the heart may represent populations that are partially differentiated, resulting in a phenotype somewhere between the SAN and the working myocardium (although, instead, it may be that primary pacemaker cells are 'specialised' versions of spontaneously beating embryonic cells, which have differentiated to develop the robustness, redundancy and regulatory feedback needed to support fast and stable pacemaking). Ectopic (Bastiaenen et al., 2012; Bredeloux et al., 2021; Nguyen et al., 2017) or enhanced (Alasti et al., 2020; Gianni, et al., 2018) automaticity of myocardium remote to the SAN may also occur in cardiac disease states (in some cases as a result of re-activation of embryonic signalling pathways) (Fazilaty & Basler, 2023; Freire et al., 2014) and lead to dangerous arrhythmic activity, underscoring the importance of an understanding of cardiac pacemaker mechanisms for devising clinical treatments.

## Conclusions

The heart is largely an autonomous organ and its continuous beating is critical for life. The autonomy of its function is based on its intrinsic spontaneous electrical activity. As part of the special issue on Pacemaking in Multicellular Organ Systems, in this review we have provided a basic description for a general audience of the known mechanisms of automaticity driving pacemaking in the heart, to allow for comparison with other organs in which pacemaking activity is important. At the same time, we have discussed often overlooked factors critical for integrated cardiac pacemaker function, including tissue-level electrophysiological and mechanical heterogeneity and pacemaker entrainment by mechano-electric feedback, which help confer a redundancy and robustness critical for the initiation of a lifetime of heartbeats. Ultimately, a better understanding of cardiac automaticity and pacemaking, and how it relates to that seen in other organs, is important for understanding physiological cardiac function, experimental model systems in which automaticity occurs and aberrant excitation that can lead to deadly cardiac arrhythmias.

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

## Additional information

### Competing interests

The authors declare that they have no competing interests.

### Author contributions

E.M. and T.A.Q. wrote the paper and agree to be accountable for all aspects of the work in ensuring that questions related to any part of its accuracy or integrity are appropriately investigated and resolved. All persons designated as authors qualify for authorship, and all those who qualify for authorship are listed.

### Funding

The authors' work is funded by the Canadian Institutes of Health Research (PJT-190009 to TAQ); the Natural Sciences and Engineering Research Council of Canada (RGPIN-2022-03150 to TAQ); and the Heart and Stroke Foundation of Canada (G-22-0032127 to TAQ).

### Keywords

calcium-clock, diastolic depolarisation, heart rate, mechano-electric feedback, mechano-sensitivity, membrane-clock, sinoatrial node

## Supporting information

Additional supporting information can be found online in the Supporting Information section at the end of the HTML view of the article. Supporting information files available:

**Peer Review History**

