## [Peer Review History · The Journal of Physiology]

Pacemaking in the heart: a redundant and robust system of mutually entrained oscillators driving cardiac automaticity

Eilidh MacDonald and T Alexander Quinn

DOI: 10.1113/JP284757

Corresponding author(s): T Alexander Quinn (alex.quinn@dal.ca)

The following individual(s) involved in review of this submission have agreed to reveal their identity: Robert D Harvey (Referee #2)

Review Timeline:

Submission Date:	01-Dec-2025
Editorial Decision:	17-Dec-2025
Revision Received:	31-Mar-2026
Accepted:	13-Apr-2026

Senior Editor: Laura Bennet

Reviewing Editor: Bernard Drumm

Transaction Report:

Dear Alex,

Re: JP-TR-2025-284757 "Pacemaking in the heart: a redundant and robust system of mutually entrained oscillators driving cardiac automaticity" by Eilidh MacDonald and T Alexander Quinn

Thank you for submitting your manuscript to The Journal of Physiology. It has been assessed by a Reviewing Editor and by 2 expert referees and we are pleased to tell you that it is acceptable for publication following satisfactory revision.

ABSTRACT FIGURES: Authors may use The Journal's premium BioRender account to create/redraw their Abstract Figures (and any other suitable schematic figure). Information on how to access this account is here: <https://physoc.onlinelibrary.wiley.com/journal/14697793/biorender-access>.

REVISION CHECKLIST: Upload a full Response to Referees file. To create your 'Response to Referees' copy all the reports, including any comments from the Senior and Reviewing Editors, into a Microsoft Word, or similar, file and respond to each point, using font or background colour to distinguish comments and responses and upload as the required file type.

We look forward to receiving your revised submission.

Best wishes,

Laura Bennet
Senior Editor

REQUIRED ITEMS

- Please include an Abstract Figure file and an Abstract Figure legend. An appropriate figure legend, which should not exceed 150 words in length, should be included in the main manuscript file. The Abstract Figure is a piece of artwork designed to give readers an immediate understanding of the research and should summarise the main conclusions. If possible, the image should be easily 'readable' from left to right or top to bottom. It should show the physiological relevance of the manuscript so readers can assess the importance and content of its findings. Abstract Figures should not merely recapitulate other figures in the manuscript. Please try to keep the diagram as simple as possible and without superfluous information that may distract from the main conclusion(s). Abstract Figures must be provided by authors no later than the revised manuscript stage and should be uploaded as a separate file during online submission labelled as File Type 'Abstract Figure'. Please also ensure that you include the figure legend in the main article file. All Abstract Figures should be created using BioRender. Authors should use The Journal's premium BioRender account to export high-resolution images. Details on how to use and access the premium account are included as part of this email.

- Please upload separate high quality figure files via the submission form.

- Author profile(s) must be uploaded via the submission form. Authors should submit a short biography (no more than 100 words for one author or 150 words in total for two authors) and a portrait photograph of the two leading authors on the paper. These should be uploaded and clearly labelled together in a Word document with the revised version of the manuscript. Any standard image format for the photograph is acceptable, but the resolution should be at least 300 DPI and preferably more. A group photograph of all authors is also acceptable, providing the biography for the whole group does not exceed 150 words.

- Please ensure that the Article File you upload is a Word file.

EDITOR COMMENTS

Reviewing Editor:

Thank you for your submission. Both reviewers have positively commented on the readability and accessibility of your review for a broad audience to inform themselves on the principles of cardiac autorhythmicity. Both reviewers have relatively minor suggestions and pointed out some areas that require correction or revising. We would bring your attention in particular to the valid points raised by reviewer 2 on how the slope of spontaneous diastolic depolarisations while important, is perhaps not the only variant worthy of discussion in the modulation of SAN excitation frequency.

Please also see 'Required Items' above.

REFEREE COMMENTS

Referee #1:

The manuscript by MacDonald and Quinn presents a basic overview of the mechanisms responsible for cardiac automaticity, intended for a general audience as part of the special issue on Pacemaking in Multicellular Organ Systems. The manuscript is well written and provides a balanced overview of basic mechanisms responsible for cardiac pacemaking, highlighting some overlooked factors that might be important for integrated cardiac pacemaker function. I have only minor concerns and suggestions, listed below.

1. Line 152: "t inwardly rectifying" and "two-I as the outward flux" - the authors should provide full names for these currents instead.

2. Lines 186 through 185: "This mechanically-induced response ... is critical to normal pacemaker function" - the authors should provide relevant references to support this statement. Stable automaticity is observed in both unloaded isolated SAN preparations and isolated SAN cells, suggesting that, while critical for modulation of SAN pacemaking, mechanical loading is not required for spontaneous automaticity - at least, in some SAN cells within the highly heterogeneous SAN structure (PMID: 28916636; PMID: 30092494).

Furthermore: "in unloaded isolated SAN preparations beating rate is often irregular, presumably due to the failure of other pacemaker mechanisms to sustain sufficient SDD. In such cases, application of a baseline level of diastolic stretch results in a stabilisation of rhythm ()" - here as well: supporting references to specific experimental studies (rather than review articles) are required. In its present form, these statements sound highly speculative.

3. Line 209 (as in line 193 and later in lines 226 and 281) - Missing reference "()".

4. Line 219 and 220: The authors should also cite a highly relevant experimental study by the Lakatta group (PMID: 28916636) (in addition to review article by Irisawa et al. 1993) which shows that spontaneous automaticity in SAN cells can exclusively rely on either membrane (If) or calcium clocks, as well as on their different combinations.

5. Regarding this point, it is worthwhile to mention another study from the Rosen group (PMID: 22753192) demonstrating that overexpression of the Ca(2+)-stimulated adenylyl cyclase AC1 in left bundle branches, alone or in combination with HCN2, provides highly efficient biological pacing and greater sensitivity to autonomic modulation than HCN2 alone. This further supports the notion that individual components of a "triple-clock" pacemaker system might be sufficient to provide cardiac automaticity. However, it is unclear whether such pacemakers form a robust and redundant pacemaker system like the SAN. In fact, subsidiary pacemakers are characterized by a significantly slower resting heart rate, slower exertional heart rates, a prolonged post-pacing recovery time, and increased beat-to-beat oscillations in cycle length compared to the SAN. In addition, these pacemakers demonstrate a higher sensitivity to acetylcholine and overdrive pacing, as well as a reduced sensitivity to beta-adrenergic stimulation.

6. Therefore, it is questionable whether all pacemaker cells represent "not-so-specialised cells", or if this is true only for subsidiary pacemakers; it is possible that SAN cells are the only true "specialized" pacemaker cells demonstrating sufficient robustness and redundancy to support fast and stable pacemaking, with adequate physiological regulatory feedback mechanisms.

Referee #2:

I enjoyed reading this review of the pacemaker mechanisms in the heart. It has done a good job of identifying most major mechanisms contributing to automaticity of the sinoatrial node (SAN), especially as it relates to the interplay of the "membrane" and "Ca²⁺" clocks, as well as the "mechanics" clock.

I do take exception, however, with the statement that the slope of spontaneous diastolic depolarization (SDD) is the key determinant of frequency of SAN excitation (lines 110 and 111). While spontaneous depolarization is clearly important, it is not the only variable. The frequency of SAN firing can also be affected by the level of the maximum diastolic potential (MDP) as well as the level of the threshold potential at which the action potential upstroke is initiated. Modulation of these other variables by neurotransmitters and hormones, as well as antiarrhythmic agents, play an important role in regulating

pacemaking activity and therefore should be mentioned as well.

Related to this, acetylcholine can contribute to changes in firing rate by reducing the influx ions through pacemaker and Ca^{2+} channels (as alluded to in lines 167 and 168), it can also influence firing by increasing the efflux of K^{+} through acetylcholine activated K^{+} channels.

There were also a few apparent typographical errors. In line 152, should "t inwardly" be "the inwardly"? In line 154, it is unclear what is meant by "two-l". And in lines 209 and 226 it appears that there may be missing references in the empty brackets.

END OF COMMENTS

We thank the editor and reviewers for their insightful and critical review of our manuscript; their feedback has helped us to improve its overall quality and potential impact. A detailed response to each of the editor's and reviewers' comments are provided below, with each comment copied and followed by a response in red. In addition, all changes to the text are indicated in the marked copy of our revised paper. We hope these revisions address the editor's and reviewers' concerns.

EDITOR COMMENTS

Reviewing Editor:

Thank you for your submission. Both reviewers have positively commented on the readability and accessibility of your review for a broad audience to inform themselves on the principles of cardiac autorhythmicity. Both reviewers have relatively minor suggestions and pointed out some areas that require correction or revising. We would bring your attention in particular to the valid points raised by reviewer 2 on how the slope of spontaneous diastolic depolarisations while important, is perhaps not the only variant worthy of discussion in the modulation of SAN excitation frequency.

Thank you for the positive comments on our manuscript; we are happy to know that we succeeded in making it readable and accessible for a broad audience. We have addressed the minor suggestions from the reviewers' (as outlined below) and have made corrections and revisions to the text where necessary (indicated in the marked copy), including a better consideration of factors influencing SAN excitation frequency beyond the slope of spontaneous diastolic depolarisation.

REFEREE COMMENTS

Referee #1:

The manuscript by MacDonald and Quinn presents a basic overview of the mechanisms responsible for cardiac automaticity, intended for a general audience as part of the special issue on Pacemaking in Multicellular Organ Systems. The manuscript is well written and provides a balanced overview of basic mechanisms responsible for cardiac pacemaking, highlighting some overlooked factors that might be important for integrated cardiac pacemaker function. I have only minor concerns and suggestions, listed below.

Thank you for the positive assessment of our paper; we are glad it was considered well-written and to provide a balanced overview of cardiac pacemaking. Thank you also for the suggestions for improvement (which we address below) – they have helped us enhance the overall quality of the paper.

1. Line 152: "t inwardly rectifying" and "two-I as the outward flux" - the authors should provide full names for these currents instead.

We are sorry for these typographical errors – it appears some of the text was accidentally 'hidden' in our submitted manuscript, so did not appear when converted to PDF. This has now been corrected in the revised version (Lines 157-160).

2. Lines 186 through 185: "This mechanically-induced response ... is critical to normal pacemaker function" - the authors should provide relevant references to support this statement.

Stable automaticity is observed in both unloaded isolated SAN preparations and isolated SAN cells, suggesting that, while critical for modulation of SAN pacemaking, mechanical loading is not required for spontaneous automaticity - at least, in some SAN cells within the highly heterogeneous SAN structure (PMID: 28916636; PMID: 30092494).

Furthermore: "in unloaded isolated SAN preparations beating rate is often irregular, presumably due to the failure of other pacemaker mechanisms to sustain sufficient SDD. In such cases, application of a baseline level of diastolic stretch results in a stabilisation of rhythm ()" - here as well: supporting references to specific experimental studies (rather than review articles) are required. In its present form, these statements sound highly speculative.

We agree the suggestion that the mechanically-induced SAN response is "critical" to normal pacemaker function is perhaps too strong a statement based on the available evidence and have edited the text to focus on its importance for modulating the SAN pacemaking, as well as added the suggested references to the text where appropriate (Lines 200-212 and 234 in the marked copy).

Regarding the stabilisation of rhythm in irregularly beating unloaded isolated SAN preparations, this is shown for rabbit and zebrafish SAN preparation in the paper by Stoyek et al., 2022, but the in-text citation was accidentally 'hidden' in our submitted manuscript so appeared as empty parentheses when converted to PDF – we are sorry for this mistake and the citation is now included in the paper (Lines 209-210 in the marked copy).

3. Line 209 (as in line 193 and later in lines 226 and 281) - Missing reference "()".

We are sorry for these errors – it appears some of the text was accidentally 'hidden' in our submitted manuscript, so did not appear when converted to PDF. This has now been corrected in the revised version (Lines 209-210, 227, 246, and 303 in the marked copy).

4. Line 219 and 220: The authors should also cite a highly relevant experimental study by the Lakatta group (PMID: 28916636) (in addition to review article by Irisawa et al. 1993) which shows that spontaneous automaticity in SAN cells can exclusively rely on either membrane (If) or calcium clocks, as well as on their different combinations.

Thank you for this suggestion. We in fact already reference the suggested study by the Lakatta group in a later section but agree that it is relevant to this section as well, so have now included it, along with a sentence to highlight its importance (Lines 232-236 in the marked copy).

5. Regarding this point, it is worthwhile to mention another study from the Rosen group (PMID: 22753192) demonstrating that overexpression of the Ca(2+)-stimulated adenylyl cyclase AC1 in left bundle branches, alone or in combination with HCN2, provides highly efficient biological pacing and greater sensitivity to autonomic modulation than HCN2 alone. This further supports the notion that individual components of a "triple-clock" pacemaker system might be sufficient to provide cardiac automaticity. However, it is unclear whether such pacemakers form a robust and redundant pacemaker system like the SAN. In fact, subsidiary pacemakers are characterized by a significantly slower resting heart rate, slower exertional heart rates, a prolonged post-pacing recovery time, and increased beat-to-beat oscillations in cycle length compared to the SAN. In addition, these pacemakers demonstrate a higher sensitivity to acetylcholine and overdrive pacing, as well as a reduced sensitivity to beta-adrenergic stimulation.

This is an interesting point, although we do not believe that the cited study (clearly) shows that individual components of a "triple-clock" pacemaker system might be sufficient to provide cardiac automaticity. In this study, the adenoviral constructs expressing HCN2, AC1, or HCN2+AC1 were injected into left bundle branch sites. As indicated, these sites are locations of 'subsidiary pacemakers' (characterised by significantly slower rates of automaticity compared to the SAN), so do not entirely lack the molecular mechanisms for pacemaking. While it is exciting to see that the inclusion of addition of pacemaking 'machinery' (calcium-stimulated adenylyl cyclase AC1 and/or HCN2) confers improved pacemaking capabilities at these sites (and may represent a novel gene therapy approach for atrioventricular block), it seems likely that these transfected pacemaking mechanisms are working in concert with mechanisms already present in the infected cells (so not 'individually').

6. Therefore, it is questionable whether all pacemaker cells represent "not-so-specialised cells", or if this is true only for subsidiary pacemakers; it is possible that SAN cells are the only true "specialized" pacemaker cells demonstrating sufficient robustness and redundancy to support fast and stable pacemaking, with adequate physiological regulatory feedback mechanisms.

Thank you for this comment; we may have been a little too strong in our suggestion that pacemaker cells represent "not-so-specialised cells". While it may be that (some) pacemaker cells are more embryonic in nature than the more differentiated working cells of the atrium and ventricle, the primary pacemaker cells of the SAN may indeed be 'specialised' versions of the spontaneously beating cells in the embryonic heart, which have differentiated to develop the robustness, redundancy, and regulatory feedback needed to support fast and stable pacemaking. We have revised the text to represent this more nuanced view (Lines 316-332 in the marked copy).

Referee #2:

I enjoyed reading this review of the pacemaker mechanisms in the heart. It has done a good job of identifying most major mechanisms contributing to automaticity of the sinoatrial node (SAN), especially as it relates to the interplay of the "membrane" and "Ca²⁺" clocks, as well as the "mechanics" clock.

Thank you for the kind words; we are happy to hear that we did a good job of summarising the major mechanisms contributing to sinoatrial node automaticity

I do take exception, however, with the statement that the slope of spontaneous diastolic depolarization (SDD) is the key determinant of frequency of SAN excitation (lines 110 and 111). While spontaneous depolarization is clearly important, it is not the only variable. The frequency of SAN firing can also be affected by the level of the maximum diastolic potential (MDP) as well as the level of the threshold potential at which the action potential upstroke is initiated. Modulation of these other variables by neurotransmitters and hormones, as well as antiarrhythmic agents, play an important role in regulating pacemaking activity and therefore should be mentioned as well.

We apologise for this clear oversight and absolutely agree that MDP and threshold potential, as well as the rate of repolarisation, also affect the frequency of SAN firing. We have revised the text to include these other factors (Lines 112-114 and 163-165 in the marked copy).

Related to this, acetylcholine can contribute to changes in firing rate by reducing the influx ions through pacemaker and Ca²⁺ channels (as alluded to in lines 167 and 168), it can also influence firing by increasing the efflux of K⁺ through acetylcholine activated K⁺ channels.

Absolutely – thank you for pointing this out. We have revised the text to include this important consideration (Lines 176-184 in the marked copy).

There were also a few apparent typographical errors. In line 152, should "t inwardly" be "the inwardly"? In line 154, it is unclear what is meant by "two-l". And in lines 209 and 226 it appears that there may be missing references in the empty brackets.

We are sorry for these typographical errors – it appears some of the text was accidentally 'hidden' in our submitted manuscript, so did not appear when converted to PDF. This has now been corrected in the revised version (Lines 157-160, 227, and 246 in the marked copy).

Dear Dr Quinn,

Re: JP-TR-2026-284757R1 "Pacemaking in the heart: a redundant and robust system of mutually entrained oscillators driving cardiac automaticity" by Eilidh MacDonald and T Alexander Quinn

We are pleased to tell you that your paper has been accepted for publication in The Journal of Physiology.

Authors should note that it is too late at this point to offer corrections prior to proofing. Major corrections at proof stage, such as changes to figures, will be referred to the Editors for approval before they can be incorporated. Only minor changes, such as to style and consistency, should be made at proof stage. Changes that need to be made after proof stage will usually require a formal correction notice.

Yours sincerely,

Laura Bennet
Senior Editor
The Journal of Physiology

P.S. - You can help your research get the attention it deserves! Check out Wiley's free Promotion Guide for best-practice recommendations for promoting your work at www.wileyauthors.com/eo/guide. You can learn more about Wiley Editing Services which offers professional video, design, and writing services to create shareable video abstracts, infographics, conference posters, lay summaries, and research news stories for your research at www.wileyauthors.com/eo/promotion.

• **IMPORTANT NOTICE ABOUT OPEN ACCESS:** To assist authors whose funding agencies mandate immediate public access to published research findings, The Journal of Physiology allows authors to pay an Open Access (OA) fee to have their papers made freely available immediately on publication.

The Corresponding Author will receive an email from Wiley with details on how to register or log in to Wiley Authors where you will be able to place an order.

You can check if your funder or institution has a Wiley Open Access Account here:
<https://authors.wiley.com/author-resources/Journal-Authors/open-access/author-compliance-tool.html>

EDITOR COMMENTS

Reviewing Editor:

Thank you for modifying your paper according to the reviewer comments. Well done on a potentially impactful review on this important area.